# Automotive Subzero Cold-Start Quasi-Adiabatic Proton Exchange Membrane Fuel Cell Fixture: Design and Validation

**DOI:** 10.3390/molecules25061410

**Published:** 2020-03-19

**Authors:** Antonio O. Pistono, Cynthia A. Rice

**Affiliations:** Department of Chemical Engineering, Tennessee Technological University, Cookeville, TN 38505, USA; aopistono21@gmail.com

**Keywords:** proton exchange membrane fuel cells, subzero cold-starts, automotive, isothermal water fill tests

## Abstract

Subzero automotive cold-starts of proton exchange membrane fuel cell (PEMFC) stacks require accelerated thermal rises to achieve nominal operating conditions and close-to-instantaneous usable output power. Advances in the material, structure and operational dependence on the balance between the maximum power output and the electrochemical conversion of hydrogen and oxygen into water requires validation with subzero cold-starts. Herein are presented the design and validation of a quasi-adiabatic PEMFC to enable single-cell evaluation, which would provide a more cost-effective option than stack-level testing. At –20 °C, the operational dependence of the preconditioned water content (3.2 verse 6.2) for a galvanic cold-start (<600 mA cm^−2^) was counter to that of a laboratory-scale isothermal water fill test (10 mA cm^−2^). The higher water content resulted in a faster startup to appreciable power output within 0.39 min versus 0.65 min. The water storage capacity, as determined from the isothermal water fill test, was greater, for the lower initial water content of 3.2, than 6.2, 17.4 ± 0.3 mg versus 12.8 ± 0.4 mg, respectively. Potentiostatic cold-starts produced usable power in 0.09 min. The versatility and reproducibility of the single cell quasi-adiabatic fixture avail it to future universal cold-start stack relevant analyzes involving operational parameters and advanced materials, including: applied load, preconditioning, interchanging flow field structures, diffusion media, and catalyst coated membranes.

## 1. Introduction

Automotive proton exchange membrane fuel cell (PEMFC) stacks are required to withstand the same environmental extremes, including subzero temperatures, as the internal combustion engine. The U.S. Department of Energy’s 2020 automotive PEMFC requirements are survivability from −40 °C and cold-starts from −30 °C. A PEMFC subzero cold-start is defined as the initiation of PEMFC operation to meet the required nominal operating temperature and power. As of 2015, the −20 °C cold-start target of 0.5 min to 50% rated power has only been met for a PEMFC stack when using one-and-a-half times the targeted parasitic shutdown/start-up energy [1]. A PEMFC stack is comprised of as many as 400 non-reactive repeat units (flow fields, coolant channels, and current collectors), each encasing the membrane electrode assembly (MEA) component. These non-reactive components behave as thermal sinks, scavenging generated heat during cold-starts. Presently, a common energy-intensive strategy is to use resistive heating to cold-start a PEMFC stack [2]. At nominal operating temperatures, a fine balance is maintained between the rate of water production and evaporative removal from the PEMFC stack. The process of subzero cold-starting of a PEMFC stack is challenging in that a balance must be attained between the rate of heat generation and product-water redistribution. At subzero temperatures, product-water accumulation and ice formation result in mass transport losses that can lead to failure if the PEMFC stack does not self-heat to above 0 °C before oxygen is completely blocked from the accessible reaction sites. As cathode catalysts and catalyst layers advance and become more efficient to meet cost targets, more parasitic power is required to self-heat non-reactive components and in addition to increased material mechanical performance/durability issues due to subzero operation. Therefore, optimization and material validation require a single-cell rapid testing platform.

Subscale single PEMFC cold-starts in standard laboratory fixtures are limited by the thermal mass of the endplates. The testing of short-stacks, of 20–30 repeat units, is cost prohibitive and strongly influenced by performance losses since the endplates behave as thermal sinks. According to the literature, the dominant subzero PEMFC studies investigating the influence of material and operational parameters have been restricted to subscale single-cell freeze-thaw testing [3,4,5,6,7,8,9,10,11,12,13,14], isothermal water fill tests [15,16,17,18,19,20,21,22,23,24,25,26,27,28,29,30,31,32,33,34,35,36,37], and non-isothermal water fill tests [3,21,33,38,39,40,41,42,43,44,45,46,47]. United Technology Corporation (UTC) Power and its corporate research facility co-developed two quasi-adiabatic single PEMFC fixtures in the mid-2000s with geometric active areas of 25 cm^2^ and 320 cm^2^ [48,49,50]. The results demonstrated that it was possible under a galvanic load cold-start to replicate the center cells in a stack’s voltage and thermal profile in the quasi-adiabatic fixture. Balliet and Newman validated their two-dimensional liquid water transport cold-start model to the UTC-Power’s quasi-adiabatic PEMFC fixture performance profiles [49]. However, the inadequate structural integrity of the quasi-adiabatic PEMFC fixture limited its reusability. The present co-author, cited in references [48,50], changed her surname after these studies were published. Published stack results are minimal and limited mostly to modeling [51,52,53,54,55,56].

Subzero PEMFC cold-starts are possible due to non-frozen water found in the membrane and catalyst layers of the MEA that support proton conduction ($\sigma_{H^{+}}$) and rapid exothermic heat generation [57,58]. Ohmic heat generation contributes to cold-start performance and is dependent on the initial water concentration. A portion of water in the ionic domains of the membrane and catalyst layers remains non-frozen at subzero temperatures due to colligative and supercooling effects [59]. Liquid water is retained within the catalyst//ionomer aggregate interfaces due to attractive forces of the charged ionomer end-chain sites ($-SO_{3}^{-}$), allowing interconnected transport through the agglomerates between the membrane and catalyst layers. The vapor-saturated water content (λ) at the aggregate interfaces is typically less than 14 and reaches a maximum of 22 for liquid saturated [54,60]. The hydrogen fuel supplied to the anode catalyst layer is electrooxidized to protons and electrons, $H_{2}\to 2H^{+}+2e^{-}$. The protons are transported through the hydrated ionic domains from the anode layer through the membrane to the cathode layer. In the presence of supplied oxygen, protons and electrons recombining within the cathode catalyst layer to form product water and heat, $O_{2}+4H^{+}+4e^{-}\to 2H_{2}O$.

Herein, a single cell quasi-adiabatic PEMFC fixture was designed to be structurally engineered for reproducible subscale cold-starts. Subzero isothermal water fill tests are shown to inadequately advance the understanding of the operational impact of the initial water content ($\lambda_{initial}$) on automotive stack relevant cold-starts. At −20 °C, isothermal water fill tests under 10 mA cm^−2^ applied loads were compared to cold-starts with loads set to 600 mA cm^−2^.

## 2. Results and Discussion

### 2.1. Quasi-Adiabatic PEMFC Fixture Design

The design constraints required for a single-cell subzero cold-start PEMFC fixture are (i) thermal isolation of the flow fields and MEA from the endplates, (ii) impervious humidified gas manifolds, (iii) structural uniformity of the active area under axial load, and (iv) high electrical conductivity between the anode and cathode sides of the PEMFC through an external circuit. Material compatibility issues of the multi-layered testing PEMFC fixture are exacerbated due to frequent thermal cycling between subzero temperatures and up to 80 °C. Figure 1 shows one side of the symmetric hardware (flow fields, heater, gas and coolant manifold, insulation, and end plates) used in the quasi-adiabatic fixture. The geometric active surface area of the quasi-adiabatic fixture was scaled down from the prototypic cell size found in automotive stacks (16 versus 320 cm^2^ active area) to simplify development and conservatively minimize any expected issues with flow field deflection; however, it is expected the active area and other geometries are scalable to develop an optimized fixture.

#### 2.1.1. Humidified Gas Manifolds

The nature of material properties makes porous structures more thermally insulated, thus satisfying constraint (i) above, but failing constraint (ii) that requires humidified gas containment. Therefore, two distinct layers were required to insulate and distribute humidified gasses to and from both sides of the MEA. Several ridged, high-density plastics were considered for the humidified gas manifold, see Table 1. All the plastics presented in Table 1 have acceptable high densities (>1200 kg m^−3^) for excluding H_2_ leakage and sufficiently high temperature limits (>121 °C) for thermal stability within the operating range of a PEMFC (−40 °C ↔ 90 °C). The manifold material must be machinable and not brittle for threaded fittings to connect humidified inlet and outlets. The ideal plastic would have a high compressive strength and compressive modulus to maintain uniformed axial load across the PEMFC fixture during thermal cycling and subsequent rebuilds (constraint (iii)), while having low thermal conductivity (constraint (i)) to retain heat generated by the MEA during PEMFC operation. An additional requirement of constraint (ii) is that the manifold material would not allow water adsorption, as it would freeze, fracture the material, and cause structural failure below 0 °C. Table 1 highlights the maximum and minimum material property values for the high-density plastics considered, underlined and underlined in shaded gray box, respectively. Only materials with low water adsorption (<1%) were considered to ensure the structure integrity of the fixture. UTC-Power adiabatic fixtures used high-density polyamide-imide (Pyropel-HD) as the internal gas manifold material with the maximum compressive modulus of all the plastics from Table 1. However, due to cost and embrittlement issues that made the inlet and outlet connector junctions prone to stress breakage, it was not selected. The manifold material was selected from the remaining plastics in Table 1 by optimizing the lowest range for both thermal conductivity and water uptake. Polyvinylidene fluoride was not selected although it had the lowest water uptake (0% of lower limit in range), as the thermal conductivity was on the higher end (47.8% of lower limit in range). Polycarbonate (PC 1000) was selected as the internal gas manifold material for the fixture as it had the lowest combination of thermal conductivity (11.0% of lower limit in range) and water uptake (8.9% of lower limit in range).

#### 2.1.2. Insulation

Several porous semi-ridged materials were considered to promote retention of the heat generated by the MEA, as listed in Table 2. The density and thermal conductivity of the materials were an order of magnitude lower than that of the humidified gas manifold, thus improving the insulating properties of the material. All the materials presented in Table 2 have acceptably low densities (< 480 kg m^−3^) that allow them to be insulative and sufficiently high temperature limits (> 149 °C) that enhance thermal stability within the range of PEMFC operation (−40 °C ↔ 90 °C). However, the decreased density made the material more porous and susceptible to entrainment of condensed water from the interior of the environmental chamber. The first three materials listed (polyisocyanurate, cellular glass, and calcium silicate) are common insulators with low compressive strength, ranging from 0.2 ↔ 0.7 MPa, and low thermal conductivity, averaging around 0.043 ± 0.022 W m^−1^ K^−1^. However, their open structure leads to high water uptake that would result in structural deformation nonuniformly altering the axial load across the active cell area, thus negatively impacting sealing of the manifold and electric continuity. The subsequent materials were of two distinctly different types: synthetic nonwoven fibrous polyimide and naturally grown balsa wood. Balsa wood is a common low-cost, low-weight construction material used for applications such as aircraft construction. In each of these families of materials, the compressive strength increases with density, and the thermal conductivity adversely increases as well. The natural strength of balsa woods is due to the multilayering of primary and secondary walls, forming a randomly distributed fiber-reinforced composite that resists out-of-plane deformation [61].

The percentage of water uptake was evaluated for both Pyropel and balsa wood by immersion in water. Immersion was a severe scenario as the only source of water that could contact the insulating material would be condensed water from the environmental chamber. Water uptake was found to be 245% over the dry weight for Pyropel MD-18, Figure 2. The as-received balsa wood water uptake was 50%, compared to the initial mass. Sealing the wood surface with a thin coating of polyurethane reduced the water uptake to only 14%. Coated balsa wood was selected as the insulating material due to the combination of relatively high compressive strength, low water uptake, and low thermal conductivity.

#### 2.1.3. Uniformity of Applied Axial Load

The peripheral manifolds and insulation parts of the assembly surrounding the MEA must apply uniform axial load across the faces of the flow fields to retain gasses and ensure electrical continuity between the catalyst layers and the current collectors. To compare the quasi-adiabatic fixture’s compression uniformity across the MEA with the standard fixture used in previous studies (Figure 3), pressure paper was used instead of the membrane and catalyst layers. Note that the vertical line in Figure 3a is an artifact from the pressure paper and should be disregarded. The standard PEMFC fixture had a flow field geometry identical to that of the quasi-adiabatic PEMFC fixture. The compression of the quasi-adiabatic fixture (Figure 3a) was mostly uniform across the face of the flow fields, but intensity was less than that of the standard fixture (Figure 3b). The torque was not increased above 70 in⋅lbs due to concerns with deflection of the endplates compromising the contact in the center of the MEA.

### 2.2. PEMFC Testing

Use of both the quasi-adiabatic fixture and a standard fixture allowed for characterization of the MEA at normal operating conditions (both fixtures), cold-starts from −20 °C, and isothermal water fill tests at −20 °C.

#### 2.2.1. Operating Performance

In Figure 4, the conditioned PEMFC H_2_/Air polarization curves of the quasi-adiabatic and a standard PEMFC fixture for the same MEA, black and gray, respectively, are compared. The 76 mΩ cm^2^ additional resistance of the MEA in the quasi-adiabatic fixture (Figure 4a) is related to the compression issues shown in Figure 3. The cell voltages are corrected using current interrupt resistance (iR-corrected) to compensate for electronic resistance losses due to the reduced axial loading of the quasi-adiabatic PEMFC fixture. The iR-corrected polarization profiles are nearly identical in the ohmic and mass transfer regions. The resistance of the quasi-adiabatic PEMFC fixture is 45% greater than that of the standard fixture.

#### 2.2.2. Water Fill Tests

There are five stages to a low-applied load water fill test: (i) initial supply of reactant gasses elevating the cell voltage, (ii) hydration of the interconnected ionomer domains of the MEA once the initial load is applied, (iii) maintenance of quasi-steady-state cell voltage during the attainment of maximum ionomer hydration, (iv) filling of the large non-hydrophilic pores of the cathode catalyst layer reducing O_2_ mass transport, and (v) freeze-out due to ice blockage in the cathode catalyst layer [18]. During these water fill tests, the product water was restricted from entering the diffusion media under the applied test conditions of low water-production rate, low-heat generation rate, and minimal flow rates (no convective transport of water). Under an applied load of 10 mA cm^−2^, the water storage capacity was evaluated for two different pre-conditioned $\lambda_{initial}$(3.2 and 6.2) at −20 °C. The $\lambda_{initial}$ was selected to match multiple published water fill tests—the lower setpoint ($\lambda_{initial}$ = 3.2) represents a dry PEMFC scenario, while the higher setpoint ($\lambda_{initial}$ = 6.2) is closer to an operating PEMFC. Both $\lambda_{initial}$ condition profiles had an initial jump in voltage although the high $\lambda_{initial}$ resulted in a higher initial cell voltage (Figure 5a) due to a lower initial resistance (Figure 5b). The maximum cell voltage was reached at similar times (>1 min) and values (0.81 V). The run with the lower $\lambda_{initial}$ stayed in the quasi-steady-state cell voltage stage >5 min longer due to a high ionomer fill capacity caused by starting at a lower ionomer hydration level and water movement from the ionomer caused by resistive heating [15]. The freeze-out stage was identical for both pre-conditioning hydration levels. The water storage capacity, calculated using Faraday’s Law, of the higher $\lambda_{initial}$ of 6.2 preconditioned water fill tests was only 12.8 ± 0.4 mg while that of the $\lambda_{initial}$ runs of 3.2 was 17.4 ± 0.3 mg.

#### 2.2.3. Cold-Starts

The impact of adjacent cell heating, $\lambda_{initial}$, and galvanic versus potentiostat applied loads were investigated on −20 °C cold-starts. The galvanically applied load of 600 mA cm^−2^ was selected to match the work published by Balliet and Newman on UTRC’s quasi-adiabatic PEMFC fixture [49]. The applied galvanic load establishes the PEMFCs maximum attainable current density with the cell voltage approaching 0 V. In a PEMFC stack, it is common for the end cells to reach negative voltages during the first few seconds of a subzero cold-start because the overall voltage of the stack is positive. Table 3 summarizes the cold-start conditions investigated within this study: $\lambda_{initial}$, adjacent cell heat-adjustment factor, and applied load. The heat-adjustment factor was included to supplement heat that would be provided from neighboring cells in a stack [49,51], as well as thermal losses in the quasi-adiabatic fixture. The heat comes from resistive heating pads located adjacent to the coolant loops of the flow fields enshrouded by the balsa wood. Each type of cold-start was preformed twice to ensure reproducibility. Representative cold-start profiles are shown in Figure 6.

Figure 6a,b compare the power density and cathode flow channel temperature versus time, respectively, for the four types of cold-start presented herein. The power densities were initially low for all of the galvanic cold-starts while the potentiostatic start instantaneously had power because the cell voltage was maintained above 0 V. For all cold-start conditions reported herein, the current density increased with time on load at subzero temperatures until the set point of 600 mA cm^−2^ could be supported by the cell voltage. Once the current density exceeded the set point, it was adjusted back down by the Scribner fuel cell software, allowing the cell voltage to rise to higher values. The current density improves as the temperature of the MEA increases, due to increased reaction kinetics and proton conduction through the ionomer. To correlate the subzero dependent current density and cell voltage response during a cold-start, the time scale origin was positioned such that it corresponded with the time the cell temperature reached 0 °C, as shown in Figure 6c. The cell-resistance profiles, proportional to proton conduction, were similar for all the cold-starts (Figure 6d) due to similar ionomer water contents, with the exception of the lower adjacent cell heating adjustment factor of 1×.

#### 2.2.4. Heat Adjustment Factor

In Figure 6, doubling the predicted adjacent cell heating adjustment factor (1× → 2×) significantly impacted the cold-start response profile for an $\lambda_{initial}$ of 3.2 is shown. The output of the heating pads (Q in Watts) was scaled by a multiplier of either 1 or 2 to equation 1 to compensate for the load-dependent fraction of heat that would be lost under applied load (current (i) in Amps) times the overpotential calculated from the difference between the thermoneutral voltage (1.48 V) and the cell voltage (V in volts) [49].
(1)Q=Ai(1.48−V)
where A is the geometric surface area of the PEMFC.

The resulting key cold-start performance metrics are summarized in Table 3 and include the average of both runs with standard deviation. The initial applied current density increased by a factor of 3.75 for the higher heating adjustment factor. The non-zero rise in voltage for the 1× heating adjustment factor occurred at a cathode flow field channel temperature around −3.1 °C, while for 2×, the transition was near 4 °C. The thermocouple point of contact is unknown within the cathode flow field channel and most likely a combination of the solid flow field temperature and the exterior of the cathode diffusion media. The mass of the flow field channels acts as a heat sink, reducing the internal temperature of the cathode flow channel, and hence, yielding the negative non-zero temperature transition for the lower heating adjustment factor. The increased heat adjustment factor suggests that a lower non-reactive thermal mass and lower thermal mass would yield a more successful cold-start. The sluggish heating profile of the lower heating adjustment factor increased the required cold-start time until usable power was available. After 1 min into the cold-start, the lower heating adjustment factor (1×) power density output was only 35% compared to the 2× cold-start (Figure 6a).

Other heating adjustment factors could be used to match other cell designs and materials. The heating pads output is an independent variable enabling the simulation of inner stack cells (symmetric heating case) or end cells (asymmetric heating case). Anomalous cells, arising from partially blocked coolant/flow field channels or degraded materials, can also give rise to other asymmetric heating cases for contiguous cells. Validation of heating adjustment factors with stack data requires significant resources and is not trivial because in-situ heating fluxes for all cells in a stack due to heterogeneous components and locally variable heat fluxes. Even if heating adjustment factors are empirically matched using single cell and stack data, a significant amount of work and resources are still required. For these reasons, validation of the quasi-adiabatic fixture with stack data was deemed outside the scope of this report.

#### 2.2.5. Initial Water Content

Increasing the $\lambda_{initial}$ from 3.2 to 6.2, using a heating adjustment factor of 2×, improved the cold-start performance; however, this result was counter to the two-thirds higher isothermal water storage capacity results (Figure 5) for the lower $\lambda_{initial}$ 3.2. For the higher $\lambda_{initial}$ found upon initially applying the load, the cell voltage could sustain nearly double current density because of the more optimal distribution of interconnected non-frozen water domains within the ionomer previously quantified with subzero electrochemical impedance spectroscopy in Dr. Rice’s lab [15]. The measured cold-start time until the voltage increased on average was reduced from 0.65 min to 0.39 min for the higher $\lambda_{initial}$, translating to appreciable power densities sooner.

#### 2.2.6. Applied Load

The type of applied load controlled the onset of appreciable power densities during the initial phase of the cold-start. Jiang and Wang demonstrated that potentiostatic cold-starts maximized the heat output [47]. The potentiostatic hold of 0.1 V multiplied the current density to get instantaneous power densities. The initial current density was 1.8× greater than that of the galvanic applied load under identical conditions.

## 3. Materials and Methods

### 3.1. PEMFC Assembly

Tests were performed in two different symmetric subscale PEMFC fixtures with an active area of 16 cm^2^. The proton-conducting membrane used was Nafion HP (Ion Power, New Castle, DE, USA). The anode and cathode catalyst layers were directly sprayed (Badger Airbrush 150) onto the membrane with a final loading of ~ 0.4 mg_Pt_ cm^−2^ (46.6% Pt on high surface area carbon, Tanaka, Chiyoda-ku, Tokyo) and 30 wt% Nafion (1100EW, Ion Power). The microporous side of the hydrophobic gas diffusion layers (SGL25BC, Ion Power) were positioned adjacent to the catalyst layers. The symmetrically sandwiched gas diffusion layers and catalyst layers around the membrane comprise the membrane electrode assembly (MEA). Polytetrafluoroethylene films (Interplast) sealed the perimeter of the compressed gas diffusion layers against the flow fields.

Figure 1 shows one side of the symmetric hardware (flow field, heater, gas and coolant manifold, insulation, and end plates) used in the quasi-adiabatic fixture. The build layup and dimensions are summarized in Table 4. Two dual-sided flow fields/coolant channels were machined out of graphite (BMC-940, MetroMold, Rogers, MN, USA), the flow fields were comprised of opposing triple serpentine channels (width 0.75 mm, depth 1 mm, and land/channel ratio 1.5) to provide reactant transport, and parallel coolant channels (width 2.54 mm, depth 1.52 mm, and land/channel ratio 1) assisted thermal management through heat generated at nominal operating temperatures of the applied load (80 °C circulating 60% ethylene glycol/40% water, Isotemp 9500, Fisher Scientific, Hampton, NH, USA).

The standard portions of the fuel-cell fixture used in both the standard and quasi-adiabatic fixtures had current collectors (gold-plated copper, electroplated in-house) compressing the flow fields. To maintain electrical continuity between the flow fields and the current collector at the non-reactive interfaces, a compressed non-hydrophobic SGL25AA was placed in the window of the polytetrafluoroethylene seal. Kapton-encased resistive heating arrays (Omega Engineering, KH-608/5-P, Norwalk, CT, USA) were positioned near the coolant side of the flow fields. Aluminum end plates (6061-T6) external to the quasi-adiabatic portion of the fixture and stainless-steel bolts torqued to 40 in-lbs. were used to maintain uniform electrical contact and force across the MEA. The quasi-adiabatic portion of the fixture had internal gas manifolds made of polycarbonate (Quadrant EPP PC 1000, Reading, PA, USA) and were insulated from the aluminum endplates with spray-polyurethane sealed Balsa wood (Specialized Balsa Wood, LLC, Loveland, CO, USA).

### 3.2. Instrumentation

A Scribner 850e fuel cell test system was the central control unit for the PEMFC testing presented herein. The system monitored cell voltage, temperature and high frequency cell resistance, while establishing reactant gas flow with specific relative humidities (RH), applied load, and isothermal temperature. A Labview program and supporting hardware were used to monitor test station/software communication and perform the necessary actions for the freezing and cold-start sequence. The thermocouple used to monitor the PEMFC temperature was a flexible ultra-fine (insulated 0.24 mm diameter) designed for in-vivo applications (T-type, Physitemp IT-24P) with an accuracy of ± 0.1 °C and located in the cathode flow field channel. Sub-zero temperatures were established using the Isotemp 9500 lab chiller and a ScienTemp 43–1.7 chest freezer equipped with a bulkhead fitting to allow electrical and feed/exit line connections. The membrane resistance was monitored under non-applied load conditions using a Milliohm meter (Agilent Technologies, 4338B, Santa Clara, CA, USA). During cold-starts, the Labview program monitored the current and voltage measured by the test station to emulate adjacent cell heating. The heating pad output was set to be a multiple of the heat that would be generated from adjacent cells in a stack (1× and 2×) and was controlled by two independent-phase angle fired controllers (Eurotherm Corp., Model-984, Worthing, United Kingdom).

### 3.3. Materials Characterization

Water uptake tests were performed on both the manifold and insulation materials by immersing approximately 10 g cubic samples in water at room temperature for 15 min and evaluating mass increases. Contact uniformity under axial load was evaluated using compression paper (super low, Fujifilm, 0.5–2.5 MPa) instead of the membrane and catalyst layers between the flow fields.

### 3.4. PEMFC BOL Conditioning

At the beginning of life (BOL), to hydrate and activate the PEMFC, 10 cathode potential cycles were run at 80 °C (gas feed dew points 75% RH) by maintaining the anode potential at 0 V vs. DHE (100% H_2_, 0.75 slpm) and varying the cathode potential by switching between 100% N_2_ (~0.12 V, 1.5 slpm) and air (>0.9 V, 21% O_2_ in a N_2_ balance, 1.5 slpm). The RH of the PEMFC feed steams was calculated from the due points (T_DP_) of the saturators for the specific cell temperature (T) according to the August-Roche-Magnus approximation [62] (Equation (2)):(2)$$\%RH=100\%\left(\frac{exp^{\left(\frac{17.625T_{DP}}{T_{DP}+243.04}\right)}}{exp^{\left(\frac{17.625T}{T+243.04}\right)}}\right)$$

Then, H_2_/Air polarization curves were performed from open circuit to 0.3 V until the voltage response profile stabilized. Between all polarization curves, the accumulated surface oxides on the cathode surface were reduced in the presence of N_2_ to remove surface oxides.

### 3.5. Freeze Pre-Conditioning

The initial water content ($\lambda_{initial}$) was reestablished prior to each subzero test by (i) repeating five H_2_/Air and H_2_/N_2_ potential cycles at 80 °C (45% RH), (ii) two H_2_/Air polarization curves at 80 °C (45% RH), and (iii) establishing the equilibrium $\lambda_{initial}$ by purging the cell with symmetric N_2_ (0.75 slpm) at either 45 °C (45% RH) or 35 °C (75% RH) for >18 h. After the equilibrium purge, the gas feed/exit lines were closed, an electrical shorting strap was placed across the anode and cathode to protect the cathode from high carbon corrosion potentials (>0.6 V), and the cell was frozen to −20 °C. The $\lambda_{initial}$’s were calculated from the feed %RH’s using the equation developed by Hinatsu et al. [63] (Equation (3)):(3)$$\lambda_{initial}=14.1{\left(\frac{\%RH}{100}\right)}^{3}-16{\left(\frac{\%RH}{100}\right)}^{2}+10.8\left(\frac{\%RH}{100}\right)+0.3$$

The $\lambda_{initial}$ values used within this study were 3.2 and 6.2 (45% RH and 75% RH, respectively). Prior to all subzero testing, the coolant was purged from the PEMFC coolant channels.

### 3.6. Water Fill Tests

After the completion of all cold-start variations, the MEA was removed from the quasi-adiabatic fixture and installed in a reference PEMFC fixture, and then subzero isothermal water fill tests were performed at −20 °C [15]. Initially, the open circuit voltage was established in the presence of H_2_/Air (0.05/0.10, 0% RH), on the anode and cathode, respectively. A small constant load of 10 mA cm^−2^ was applied until the cell voltage dropped below 0.1 V. Runs were repeated 2–3 times to ensure accuracy.

### 3.7. Cold-Starts

Subzero cold-starts were performed at −20 °C in the quasi-adiabatic PEMFC fixture. Initially, the open-circuit voltage was established in the presence of H_2_/Air (0.5 slpm/0.75 slpm, 0% RH), on the anode and cathode, respectively. Under applied load using the upper set point value of 600 mA cm^−2^, the stoichiometry was never less than 2. The applied load was controlled either galvanically or potentiostatically. The galvanic loads were ramped up to the set point in less than 1 min as the non-negative PEMFC voltage could sustain higher currents. The potentiostatic hold was initially set to 0.1 V. The output of the heating pads (Q in Watts) was scaled by a multiplier of either 1× or 2× to Equation (1). Runs were repeated 2–3 times to ensure accuracy.

## 4. Conclusions

Single-cell, −20 °C cold-starts were attained in a quasi-adiabatic fixture, consisting of polycarbonate gas manifolds and balsa wood insulation. This fixture used heating pads placed on the exterior of the internal flow fields to simulate the anticipated heat from adjacent cells in a stack. A 2× heating factor was used due to adjacent cell heating and thermal losses from the flow-field mass. The quasi-adiabatic single-cell fixture can emulate the thermal temperature rise and product water redistribution during cold-starts. Only a limited number of published, stack-level cold-start results, restricted mostly to simulations, are presented in the literature. The majority of the published subzero PEMFC testing is done on single cells and quantifies the water fill capacity before freezeout using a water fill test. The results presented herein succinctly demonstrate the inadequacies of the commonly used lab scale isothermal water fill tests in validating operational and material subzero cold-start capabilities. The higher rate of water production during the galvanic cold-starts (600 mA cm^−2^) showed maximum hydration of the membrane within less than 2 min in contrast to the 4–8 min required in the isothermal water fill test (10 mA cm^−2^). As the internal cell temperature rose above 0 °C during a cold-start, nearly 20 mg of water were produced. However, for the isothermal water fill test, the highest water fill capacity ($\lambda_{initial}$ = 3.2) was only 17.4 mg. The higher $\lambda_{initial}$ of 6.2 had a lower isothermal water storage capacity than that of 3.2, but conversely, a galvanic cold-start resulted in a shortened time to usable power. The potentiostatic cold-start (0.1 V) provided useful power immediately, resulting in superior cold-start performance.

## Figures and Tables

**Figure 1 molecules-25-01410-f001:**
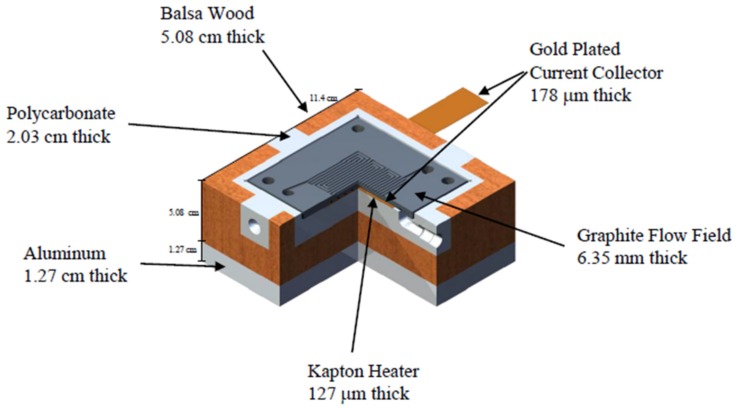
One half of symmetric quasi-adiabatic single-cell hardware. The parallel coolant channels are machined into the backside of the graphite flow field plate.

**Figure 2 molecules-25-01410-f002:**
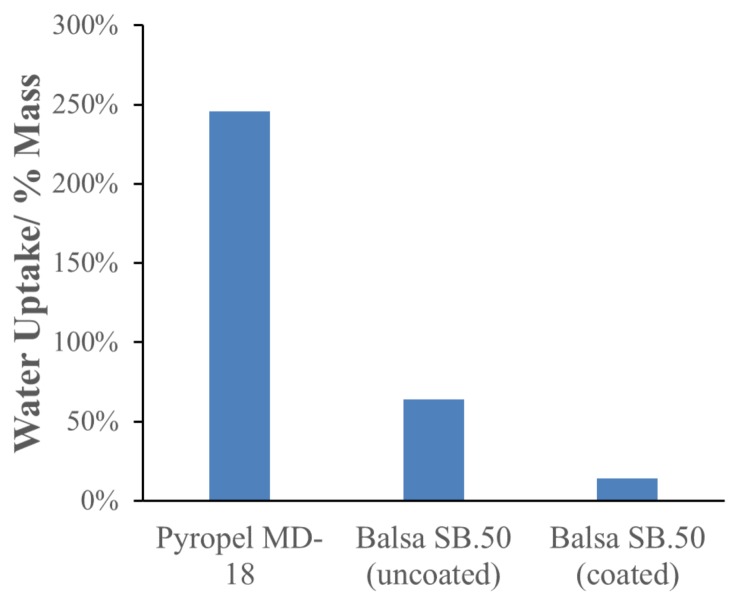
Water uptake of down-selected insulation materials for 15-min water immersion.

**Figure 3 molecules-25-01410-f003:**
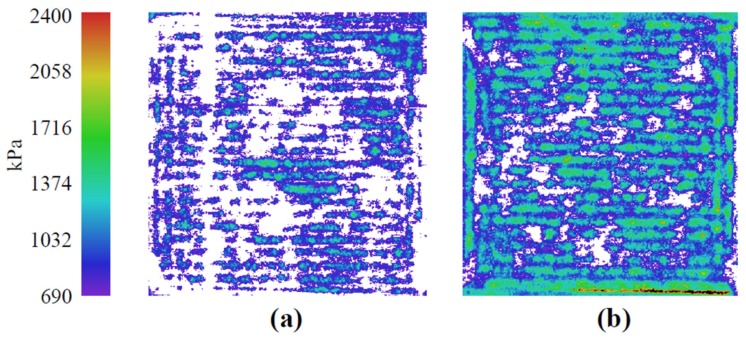
Compression paper for contact intensity and uniformity between the 16 cm^2^ flow fields (**a**) quasi-adiabatic fixture and (**b**) standard fixture. Each fixture was torqued to 70 in⋅lbs.

**Figure 4 molecules-25-01410-f004:**
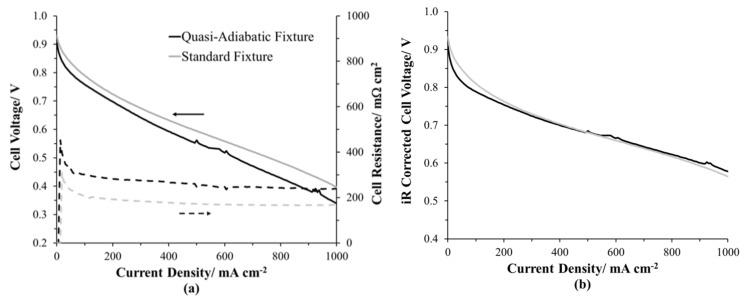
H_2_/21% O_2_ polarization curves at 80 °C cell temperature and 44.3% relative humidity under ambient pressure for the quasi-adiabatic fixture and a standard fixture (**a**) cell voltage and (**b**) compensated iR free cell voltage.

**Figure 5 molecules-25-01410-f005:**
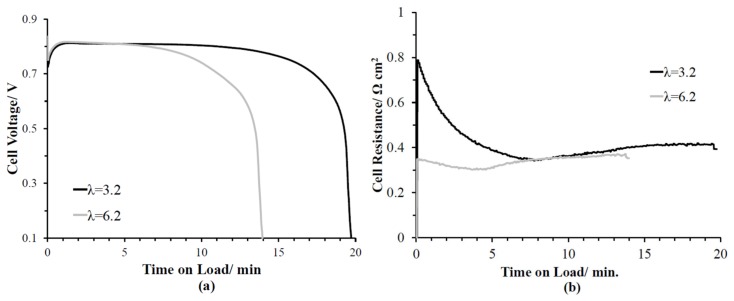
Preconditioned initial water content runs (3.2 and 6.2) isothermal water fill test at 10 mA cm^−2^, −20 °C and H_2_/21% O_2_ (0.05/0.1 lpm, respectively). (**a**) Cell voltage and (**b**) cell resistance versus time on load.

**Figure 6 molecules-25-01410-f006:**
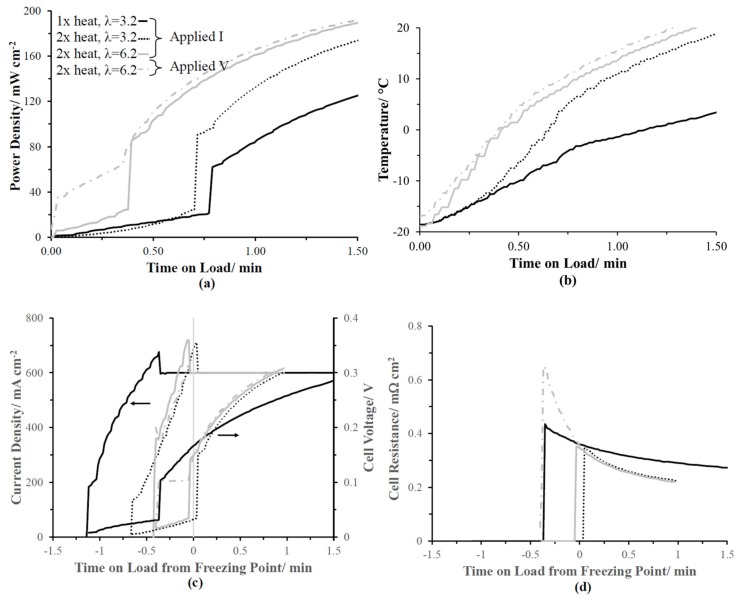
Preconditioned initial water content runs (3.2 and 6.2), cold-start tests under set galvanic load of 600 mA cm^−2^, or applied potentiostatic hold of 0.1 V at −20 °C and H_2_/21% O_2_ (0.5/0.75 lpm, respectively). (**a**) Power density and (**b**) cathode channel temperature versus time on load. (**c**) Current density and (**d**) cell resistance vs. time on load from 0 °C.

**Table 1 molecules-25-01410-t001:** High-density polymers considered for the gas manifold of the quasi-adiabatic PEMFC fixture. The highest and lowest material properties values of each type are underlined and the highest is further identified by a shaded gray box. Not available (N/A).

Polymer Type	Trade Name	Supplier	Density (kg m^−3^)	Maximum Operating (°C)	Compressive Strength (MPa)	Compressive Modulus (GPa)	Thermal Conductivity (W m^−1^ K^−1^)	Water Uptake (%)
Polyvinylidene fluoride	Symalit unfilled	Quadrant	***1780***	149	***68.9*** ***(10% def)***	***1.1***	0.216	***0.05***
Polyimide	D7000 PI	Quadrant	1380	240	***145*** ***(5% nom strain)***	N/A	0.22	***4***
Polyethersulfone	PES	Westlake Plastic	1370	N/A	100	2.68	0.239	1.85
Polyamide-imide	Pyropel-HD	Albany International	1360	***288***	N/A	***3.72***	0.23	Est 0.58
Polyaryletherketone	AV-848	Solvay AvaSpire	1320	N/A	118	N/A	0.22	0.5
Polyetherimide	Duratron U1000	Quadrant	1280	171	152 (10% def)	3.31	***0.177***	1.25
Polysulfone	PSU 1000	Quadrant	1240	149	89.6 (10% def)	2.59	***0.259***	0.6
Polycarbonate	PC 1000	Quadrant	***1200***	***121***	79.3 (10% def)	2.07	0.186	0.4

**Table 2 molecules-25-01410-t002:** Semi-ridged materials considered as insulation of the quasi-adiabatic PEMFC fixture. The highest and lowest material properties values of each type are underlined and the highest is further identified by a shaded gray box. Not available (N/A).

Material	Trade Name	Supplier	Density (kg m^−3^)	Maximum Operating (°C)	Compressive Strength (MPa)	Compressive Modulus (GPa)	Thermal Conductivity (W m^−1^ K^−1^)
Polyisocyanurate	TRYMER 2000 XP	ITW Insulation Systems	***32.8***	**149**	0.16«0.21	***0.003 << 0.005***	***0.027***
Cellular Glass	FOAMGLAS ONE	Pittsburg Corning	117	482	0.62	0.9	0.032«0.054
Calcium Silicate	Thermo-12 Gold	Johns Manville	230	***1200***	0.690 (5% def)	N/A	0.053«0.058
Nonwoven Polyimide	Pyropel MD-12	Albany International	190	288	***0.07***	0.006	0.036
Nonwoven Polyimide	Pyropel MD-18	Albany International	290	288	0.1	0.015	0.041
Nonwoven Polyimide	Pyropel MD-30	Albany International	***480***	288	0.41	0.1	0.049
Balsa	SB.50	AIREX AG BALTEK	109	163	5.5	1.6	0.048
Balsa	SB.100	AIREX AG BALTEK	148	163	9.2	2.5	0.066
Balsa	SB.150	AIREX AG BALTEK	285	163	***22***	***4.4***	***0.084***

**Table 3 molecules-25-01410-t003:** Cold-start parameters used in tests from Figure 6 and corresponding select performance metrics.

Cold-Start Parameters				
Heat Adjustment Factor	1×	2×	2×	2×
Initial Water Content	3.2	3.2	6.2	6.2
Applied Load	600 mA cm^−2^	600 mA cm^−2^	600 mA cm^−2^	0.1 V
**Cold-Start Performance Metrics**				
Initial Current Density (mA cm^−2^)	47 ± 15	129 ± 8	361 ± 3	402 ± 3
Temperature of Cell Voltage Rise (°C)	−3.1 ± 0.1	4.0 ± 0.6	−0.9 ± 1.0	−1.8 ± 1.0
Time to Cell Voltage Rise (min)	0.78 ± 0.01	0.65 ± 0.09	0.39 ± 0.01	0.38 ± 0.01
Time to Power > 40 mA cm^−2^ (min)	0.78 ± 0.01	0.65 ± 0.09	0.39 ± 0.01	0.09 ± 0.01
Time to 20 °C (min)	5.6 ± 0.1	1.6 ± 0.1	1.4 ± 0.1	1.4 ± 0.1

**Table 4 molecules-25-01410-t004:** Symmetric lay-up of quasi-adiabatic PEMFC fixture centered around the proton-conducting membrane.

	PEMFC Fixture Component	Count	Thickness (cm)	Total Area (cm^2^)	Seal (cm)
Standard Components	Membrane Electrode Assembly	Membrane (center of build)	1	0.002	20.25	
Catalyst Layer	2	> 0.001	16	
Gas Diffusion Layer (uncompressed)	2	0.0235	16	0.0152
Flow Field	2	0.635	62.4	
Electrical Contactor (uncompressed)	2	0.019	16	0.0102
Current Collector	2	0.0178	25.8	
Adiabatic Portion	Heating Pad	2	0.0127	25.8	
Manifold	2	2.03	84.5	
Insulator	2	5.08	130.6	
Endplate	2	1.27	130.6

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
