# Peer review of "Automotive Subzero Cold-Start Quasi-Adiabatic Proton Exchange Membrane Fuel Cell Fixture: Design and Validation"

_molecules, 2020, doi:10.3390/molecules25061410_

Round 1
Reviewer 1 Report
Paper presents a new adiabatic cell for coldstart studies. The paper should be accepted for publication with minor edits
Was potentiostatic coldstart attempted for lamda = 3.2
Page 1 line 30, 31,32: degree symbol missing
Line 23: switching between the requirements in s and minutes. Keep the units consistent.
Line 77 missing content in the bracket
Page 9 line 232 figure number missing
Figure 6 b the line not distinguishable between the 1x and 2x for lama = 3.2
Author Response
Reviewer 1:
Paper presents a new adiabatic cell for coldstart studies. The paper should be accepted for publication with minor edits
- Was potentiostatic coldstart attempted for lamda = 3.2
Response: The Authors appreciate the Reviewers comment; however, this specific parameter was not evaluated. The reason was this particular parameter was not evaluated for poteniostatic subzero cold-starts was the effect of lambda was probed for the constant load subzero cold-starts. The potentiostatic subzero cold-start was only probed for the highest performing conditions, i.e. an initial water content of 6.2 and adjacent heating power of 2x. There was no reason to expect the performance would not follow the initial water content and adjacent heating power trends.
- Page 1 line 30, 31,32: degree symbol missing
Response: The Authors thank the Reviewer for identifying this formatting issue. It has been corrected.
- Line 23: switching between the requirements in s and minutes. Keep the units consistent.
Response: The Authors thank the Reviewer for identifying this issue. All time units have been converted to minutes.
- Line 77 missing content in the bracket
Response: The Authors thank the Reviewer for identifying this formatting issue. It has been corrected.
- Page 9 line 232 figure number missing
Response: The Authors thank the Reviewer for identifying this issue. It has been corrected to say Figure 5.
- Figure 6 b the line not distinguishable between the 1x and 2x for lama = 3.2
Response: The Authors thank the Reviewer for identifying this formatting issue. It has been corrected.

Reviewer 2 Report
The manuscript by Pistono and Rice described the development of a PEMFC single cell specially designed to mimic cold start in a PEMFC stack. Thereby boundary conditions present in a stack (heat exchange) are theoretically considered to make a selection of materials. The paper is focussed on the design and engineering of the test cell itself; it does not provide additional insight into the cold-start processes. The paper is well written and the presentation of the results and experimental details is adequate. However, there are some issues:
1) the used flow field is a serpentine flow field in a small size lab cell of 16cm2 which is not representative for stack conditions. The authors should comment on that.
2) the cell seem to still face some issue with compression (clamping pressure) which is the reason why iR-corrected polarization curves are presented to eliminate ohmic/contact resistance losses. To assess the performance of the new cell it is required to provide non iR-corrected pol curves as well.
3) The author's motivation is to provide a cost efficient single cell testing platform for cold stat instead of using stack-level testing. However, it is not properly discussed if this new test platform provides comparable results as in stack. This is crucial to asses its suitability. This comparison should be clear and this should be one of the main conclusions addressed both in the Abstract and in the Conclusion section.
4) the Conclusions section provides a summary of the conducted work but lacks real conclusions. The section has to be modified so it provides real conclusions highlighting what are the advantages of the adiabatic fixture cell compared to other cells.
Taking into account the engineering character of the paper (little scieitifc novelty) focussing on designing of a very specific PEMFC test cell for cold-start and the lack of clear conclusions, it has very limited relevance of the readership of Molecules. Therefore, my recommendation is: reject for publication in Molecules.
However, I also recommend resubmission of the paper (after carefully addressing the above mentioned issues - major revision) to a fuel cell focussed journal, such as Fuel Cells or J.Power Sources. Alternatively, a more suitable MDPI open access journal (which scope includes Thermodynamics, Hydrogen Technologies etc.) would be Energies.
Author Response
Reviewer 2:
The manuscript by Pistono and Rice described the development of a PEMFC single cell specially designed to mimic cold start in a PEMFC stack. Thereby boundary conditions present in a stack (heat exchange) are theoretically considered to make a selection of materials. The paper is focussed on the design and engineering of the test cell itself; it does not provide additional insight into the cold-start processes. The paper is well written and the presentation of the results and experimental details is adequate. However, there are some issues:
- the used flow field is a serpentine flow field in a small size lab cell of 16cm2 which is not representative for stack conditions. The authors should comment on that.
Response: The Authors thank the Reviewer for identifying this omission. The following has been added to section 2.1: “The geometric active surface area of the quasi-adiabatic fixture was scaled down from the prototypic cell size found in automotive stacks (16 versus 320 cm2 active area) to simplify development and conservatively minimize any expected issues with flow field deflection; however, it is expected the active area and other geometries are scalable to develop an optimized fixture.”
- the cell seem to still face some issue with compression (clamping pressure) which is the reason why iR-corrected polarization curves are presented to eliminate ohmic/contact resistance losses. To assess the performance of the new cell it is required to provide non iR-corrected pol curves as well.
Response: The Authors concur with the Reviewer that some minimal compression issues remain. To comply with the Reviewer’s suggestion, Figure 4 has been separated into two graphs: (a) cell voltage and resistance and (b) iR corrected cell voltage vs. current density. And the following statement had been added to the discussion “The 76 mW cm2 additional resistance of the MEA in the quasi-adiabatic fixture (Figure 4a) is related to the compression issues shown in Figure 3.”.
- The author's motivation is to provide a cost efficient single cell testing platform for cold stat instead of using stack-level testing. However, it is not properly discussed if this new test platform provides comparable results as in stack. This is crucial to asses its suitability. This comparison should be clear and this should be one of the main conclusions addressed both in the Abstract and in the Conclusion section.
Response: The Authors concur with the Reviewer that a direct comparison of stack performance to the quasi-adiabatic single fuel cell fixture is desired. However, stack testing in an academic lab is cost prohibitive and published cold-starts in the literature are limited. Those that are presented are MEA dependent. The corresponding Author’s previous work at United Technologies (ref 48 and 50) demonstrate that the fixture’s cold-start performance can be tuned to match that of a stack.
End of Abstract: “The versatility and reproducibility of the single cell quasi-adiabatic fixture avail it to future universal cold-start stack relevant analyzes involving operational parameters and advanced materials, including: applied load, preconditioning, interchanging flow field structures, diffusion media, and catalyst coated membrane.”
Line 66-67 in Introduction: “Published stack results are minimal and limited mostly to modeling[51-56].”
Section 2.2.4: “Other heating adjustment factors could be used to match other cell designs and materials. The heating pads output is an independent variable enabling the simulation of inner stack cells (symmetric heating case) or end cells (asymmetric heating case). Anomalous cells, arising from partially blocked coolant/flow field channels or degraded materials, can also give rise to other asymmetric heating cases for contiguous cells. Validation of heating adjustment factors with stack data requires significant resources and is not trivial because in-situ heating fluxes for all cells in a stack due to heterogeneous components and locally variable heat fluxes. Even if heating adjustment factors are empirically matched using single cell and stack data, a significant amount of work and resources are still required. For these reasons, validation of the quasi-adiabatic fixture with stack data was deemed outside the scope of this report.”
Conclusion: “Single-cell, -20°C cold-starts were attained in a quasi-adiabatic fixture, consisting of polycarbonate gas manifolds and balsa wood insulation. This fixture used heating pads placed on the exterior of the internal flow fields to simulate the anticipated heat from adjacent cells in a stack. A 2× heating factor was used due to thermal losses from the flow-field mass. The quasi-adiabatic single-cell fixture can emulate the thermal temperature rise and product water redistribution during cold-starts. Only a limited number of published stack-level, cold-start results, restricted mostly to simulations, are presented in the literature. The majority of the published subzero PEMFC testing is done on single cells and quantifies the water fill capacity before freezeout using a water fill test. The results presented herein succinctly demonstrate the inadequacies of the commonly used lab scale isothermal water fill tests in validating operational and material subzero cold-start capabilities. The higher rate of water production during the galvanic cold-starts (600 mA cm-2) showed maximum hydration of the membrane within less than 2 min in contrast to the 4 – 8 min required in the isothermal water fill test (10 mA cm-2). As the internal cell temperature rose above 0°C during a cold-start, nearly 20 mg of water were produced. However, for the isothermal water fill test, the highest water fill capacity (l_initial = 3.2) was only 17.4 mg. The higher l_initial of 6.2 had a lower isothermal water storage capacity than that of 3.2, but conversely, a galvanic cold-start resulted in a shortened time to usable power. The potentiostatic cold-start (0.1 V) provided useful power immediately, resulting in superior cold-start performance.”
- the Conclusions section provides a summary of the conducted work but lacks real conclusions. The section has to be modified so it provides real conclusions highlighting what are the advantages of the adiabatic fixture cell compared to other cells.
Response: The Authors graciously thank the Reviewer for identify the issues with our conclusions. We have written the conclusions see response to comment 3 above.
Taking into account the engineering character of the paper (little scieitifc novelty) focussing on designing of a very specific PEMFC test cell for cold-start and the lack of clear conclusions, it has very limited relevance of the readership of Molecules. Therefore, my recommendation is: reject for publication in Molecules.
However, I also recommend resubmission of the paper (after carefully addressing the above mentioned issues - major revision) to a fuel cell focussed journal, such as Fuel Cells or J.Power Sources. Alternatively, a more suitable MDPI open access journal (which scope includes Thermodynamics, Hydrogen Technologies etc.) would be Energies.
Reviewer 3 Report
The paper discusses an interesting experimental investigation concerning the design and performance evaluation of a quasi-adiabatic fixture of a Proton Exchange Membrane Fuel Cell that undergoes cold-start from -20°C. The analysis is performed with attention to different initial water contents in terms of start-up time and provided power.
The overall work is interesting, the material and methods are presented in detail and both context and motivations are addressed. Nevertheless, there are some points that needs attention from the Authors before reaching a suitable quality for publication.
First of all, the novelty of the proposed study with respect to the literature is not fully remarked. An initial literature overview is given, however, a proper definition of the advancements with respect to what already available on the market and addressed in the state-of-the-art is not evident. Therefore, the Authors are invited to enrich the literature overview concerning cold start-up studies, so as to better single out their contribution.
The Results and Discussion section is not clearly written. As first, it should be located after the Section 3 Materials and Methods, which should become Section 2 instead. Afterwards, the Authors should better justify and comment the results presented in Figure 4: a detailed explanation on the different performance between quasi-static and standard fixtures should be given. Indeed, it is not clear how a such great discrepancy between the cell resistances give an almost similar performance in terms of polarization curves. Moreover, the Authors should explain the significance of “iR-correction” used to compare the polarization profiles.
The heat-adjustment factor introduced the first time in section 2.2.3 is not described anywhere in the Manuscript. This does not allow a clear comprehension of the proposed results. Moreover, the Authors should comment on the choice of the cold start parameters presented in Table 3: the investigation of the four cases does not cover a uniformity of parameters values, since a proper combination of factors is not achieved (for instance, the condition of potentiostatic mode with 1x factor and 3.2 water content is not investigated).
Author Response
Reviewer 3:
The paper discusses an interesting experimental investigation concerning the design and performance evaluation of a quasi-adiabatic fixture of a Proton Exchange Membrane Fuel Cell that undergoes cold-start from -20°C. The analysis is performed with attention to different initial water contents in terms of start-up time and provided power.
The overall work is interesting, the material and methods are presented in detail and both context and motivations are addressed. Nevertheless, there are some points that needs attention from the Authors before reaching a suitable quality for publication.
- First of all, the novelty of the proposed study with respect to the literature is not fully remarked. An initial literature overview is given, however, a proper definition of the advancements with respect to what already available on the market and addressed in the state-of-the-art is not evident. Therefore, the Authors are invited to enrich the literature overview concerning cold start-up studies, so as to better single out their contribution.
Response: The Authors graciously thank the Reviewer for identify this issue. To address the issue of ‘novelty’ the following additions have been made:
End of Abstract: “The versatility and reproducibility of the single cell quasi-adiabatic fixture avail it to future universal cold-start stack relevant analyzes involving operational parameters and advanced materials, including: applied load, preconditioning, interchanging flow field structures, diffusion media, and catalyst coated membrane.”
Line 66-67 in Introduction: “Published stack results are minimal and limited mostly to modeling[51-56].”
Conclusion: “Single-cell, -20°C cold-starts were attained in a quasi-adiabatic fixture, consisting of polycarbonate gas manifolds and balsa wood insulation. This fixture used heating pads placed on the exterior of the internal flow fields to simulate the anticipated heat from adjacent cells in a stack. A 2× heating factor was used due to thermal losses from the flow-field mass. The quasi-adiabatic single-cell fixture can emulate the thermal temperature rise and product water redistribution during cold-starts. Only a limited number of published stack-level, cold-start results, restricted mostly to simulations, are presented in the literature. The majority of the published subzero PEMFC testing is done on single cells and quantifies the water fill capacity before freezeout using a water fill test. The results presented herein succinctly demonstrate the inadequacies of the commonly used lab scale isothermal water fill tests in validating operational and material subzero cold-start capabilities. The higher rate of water production during the galvanic cold-starts (600 mA cm-2) showed maximum hydration of the membrane within less than 2 min in contrast to the 4 – 8 min required in the isothermal water fill test (10 mA cm-2). As the internal cell temperature rose above 0°C during a cold-start, nearly 20 mg of water were produced. However, for the isothermal water fill test, the highest water fill capacity (l_initial = 3.2) was only 17.4 mg. The higher l_initial of 6.2 had a lower isothermal water storage capacity than that of 3.2, but conversely, a galvanic cold-start resulted in a shortened time to usable power. The potentiostatic cold-start (0.1 V) provided useful power immediately, resulting in superior cold-start performance.”
- The Results and Discussion section is not clearly written. As first, it should be located after the Section 3 Materials and Methods, which should become Section 2 instead.
Response: The Authors redirect this comment to the editor, as they were complying with the Journals formatting guidelines.
- Afterwards, the Authors should better justify and comment the results presented in Figure 4: a detailed explanation on the different performance between quasi-static and standard fixtures should be given. Indeed, it is not clear how a such great discrepancy between the cell resistances give an almost similar performance in terms of polarization curves. Moreover, the Authors should explain the significance of “iR-correction” used to compare the polarization profiles.
Response: The Authors thank the Reviewer for identifying this source of confusion. To add clarity Figure 4 has been separated into two graphs: (a) cell voltage and resistance and (b) iR corrected cell voltage vs. current density. The following statement has been added to the discussion section “The 76 mW cm2 additional resistance of the MEA in the quasi-adiabatic fixture (Figure 4a) is related to the compression issues shown in Figure 3.”
- The heat-adjustment factor introduced the first time in section 2.2.3 is not described anywhere in the Manuscript. This does not allow a clear comprehension of the proposed results.
Response: The Authors are grateful for this observation by the Reviewer and have corrected the lack of description in section 2.2.4 as follows: “The output of the heating pads (Q in Watts) was scaled by a multiplier of either 1 or 2 to equation 1 to compensate for the load-dependent fraction of heat that would be lost under applied load (current (i) in Amps) times the overpotential calculated from the difference between the thermoneutral voltage (1.48 V) and the cell voltage (V in volts) [49].
(1)
Where A is the geometric surface area of the PEMFC.”
Additionally, to section 2.2.3 the following description was added: “The heat-adjustment factor was included to supplement heat that would be provided from neighboring cells in a stack [49, 62], as well as thermal losses in the quasi-adiabatic fixture. The heat comes from resistive heating pads located adjacent to the coolant loops of the flow fields enshrouded by the balsa wood.”
- Moreover, the Authors should comment on the choice of the cold start parameters presented in Table 3: the investigation of the four cases does not cover a uniformity of parameters values, since a proper combination of factors is not achieved (for instance, the condition of potentiostatic mode with 1x factor and 3.2 water content is not investigated).
Response: The Authors concur with the Reviewer that the study presented does not cover all interconnected operational performance dependencies on cold-starts. That will be thoroughly explored in future works, guided by a statistical design of experiments to probe the degree of interdependence. The motivation of the present manuscript was to show the design of the quasi-adiabatic fixture and the impact of select parameters on cold-start performance. Regarding the potentiostatic cold-start at an initial water content of 3.2, we anticipate it to follow a similar trend in comparison to the 6.2 with the results for the galvanic cold-start.
The following parameter justification was added to section 2.2.2 “The was selected to match multiple published water fill test – the lower setpoint ( = 3.2) represents a dry PEMFC scenario, while the higher setpoint ( = 6.2) is closer to an operating PEMFC.”

Round 2
Reviewer 2 Report
The authors have addressed the reviewer's comments point by point and modified the manuscript accordingly by changing text an adding further polarization curves. The overall revision is acceptable and no further changes are suggested.
Reviewer 3 Report
The Authors properly addressed the issues remarked in the previous review process. Therefore, the paper can be now accepted for publication.